# The Unique Homothallic Mating-Type Loci of the Fungal Tree Pathogens *Chrysoporthe syzygiicola* and *Chrysoporthe zambiensis* from Africa

**DOI:** 10.3390/genes14061158

**Published:** 2023-05-26

**Authors:** Nicolaas A. van der Merwe, Tshiamo Phakalatsane, P. Markus Wilken

**Affiliations:** Department of Biochemistry, Genetics and Microbiology, Forestry and Agricultural Biotechnology Institute, University of Pretoria, Pretoria 0028, South Africa; albe.vdmerwe@fabi.up.ac.za (N.A.v.d.M.);

**Keywords:** *Chrysoporthe*, ascomycete, mating-type, homothallic

## Abstract

*Chrysoporthe syzygiicola* and *C. zambiensis* are ascomycete tree pathogens first described from Zambia, causing stem canker on *Syzygium guineense* and *Eucalyptus grandis*, respectively. The taxonomic descriptions of these two species were based on their anamorphic states, as no sexual states are known. The main purpose of this work was to use whole genome sequences to identify and define the mating-type (*MAT1*) loci of these two species. The unique *MAT1* loci for *C. zambiensis* and *C. syzygiicola* consist of the *MAT1-1-1*, *MAT1-1-2*, and *MAT1-2-1* genes, but the *MAT1-1-3* gene is absent. Genes canonically associated with opposite mating types were present at the single mating-type locus, suggesting that *C. zambiensis* and *C. syzygiicola* have homothallic mating systems.

## 1. Introduction

Ascomycete sexual mating strategies are generally categorized as either homothallic or heterothallic. Heterothallic individuals require a genetically compatible partner to complete their sexual cycle [1,2], while homothallic fungi are self-fertile and do not require a partner for sexual reproduction [3,4]. Homothallism allows fungi to produce self-fertile offspring that can quickly colonize a new niche, but some of these fungi are also able to outcross under conducive environmental conditions [5]. Heterothallism and obligate sexual outcrossing are beneficial in circumstances where mating-type partners are plentiful and fitness costs for selfing are considerable [6], for example, when genetic diversity in a population is selectively advantageous.

In ascomycetes, mating is governed by mating-type genes that are located within the *MAT1* locus [7,8]. These genes are primary regulators of reproduction, and function as determinants of mating compatibility [8,9]. In heterothallic fungi, the *MAT1* locus consists of either a *MAT1-1* or a *MAT1-2* idiomorph [3]. The *MAT1-1* idiomorph is minimally defined by the *MAT1-1-1* gene encoding a protein with an alpha-1 box, whereas the *MAT1-2* idiomorph is minimally defined by the *MAT1-2-1* gene that encodes a protein with a high-mobility-group (HMG) box domain [2,10,11]. However, in homothallic species the *MAT1* locus harbours homologous genes that are associated with both *MAT1-1* and *MAT1-2* in the same genome, i.e., both the *MAT1-1-1* and *MAT1-2-1* genes are present. The mating genes in homothallic species can either be linked in a single locus, or they can occur unlinked at separate loci in the genome [2,4].

The genus *Chrysoporthe* consists of fungal pathogens that cause Chrysoporthe canker of Myrtales trees, notable economically important forest trees as well as ornamental trees [12,13]. Most species of *Chrysoporthe* commonly display sexual fruiting bodies (perithecia) in natural habitats [13,14,15,16,17], although sexual reproduction is not frequently observed under laboratory conditions. For other *Chrysoporthe* species, such as *C. hodgesiana*, *C. zambiensis*, and *C. syzygiicola*, perithecia are rarely observed even under natural conditions [18,19]. In the absence of observable perithecia, the mating-type genes can provide evidence for the possibility of sexual reproduction in these species.

The mating-type loci of *C. austroafricana*, *C. cubensis*, and *C. deuterocubensis* have previously been characterized [20]. For example, *C. austroafricana* has a heterothallic mating system that consists of either a *MAT1-1* or a *MAT1-2* idiomorph in a single haploid genome. The *MAT1-1* idiomorph of *C. austroafricana* contains the *MAT1-1-1*, *MAT1-1-2*, and *MAT1-1-3* genes. Therefore, the genetic composition of the *MAT1-1* idiomorph of *C. austroafricana* is similar to that of other heterothallic species in the Sordariomycetes [21,22,23]. The *MAT1-2* idiomorph of *C. austroafricana* contains the *MAT1-2-1* gene, but also truncated versions of the *MAT1-1-1* and *MAT1-1-2* genes that are usually associated with the *MAT1-1* idiomorph [20]. Thus, the mating-type idiomorphs of *C. austroafricana* have unique and distinctive organization and gene content. On the other hand, the *MAT1* loci of *C. cubensis* and *C. deuterocubensis* are typical for homothallic species.

Previous research [20] has been important in determining the mating systems of three *Chrysoporthe* species that occur in Africa. However, there is little genetic information regarding the mating-type configurations of other *Chrysoporthe* species. Population studies have attempted to infer mating systems in some *Chrysoporthe* species by considering genetic diversity. An example is *Chrysoporthe puriensis* from Brazil, for which microsatellite markers were used to reveal a high level of genetic diversity [24]. Such high levels of diversity might be an indicator of recombination, but cannot be used to infer the mating system, since both homothallic and heterothallic species can outcross.

The mode and genetic basis of sexual reproduction in *C. zambiensis* and *C. syzygiicola*, both African species, are unknown. Therefore, the aim of this study was to characterize the mating-type genes of these species, and infer their mating systems. To accomplish this goal, whole genome sequencing was used to enable gene identification and characterization. Phylogenies were constructed to investigate any conflicts that might exist between the *MAT1* genes and the species phylogeny.

## 2. Materials and Methods

### 2.1. Genome Sequencing, Assembly, and Analysis

*Chrysoporthe syzygiicola*: Zambia, Luapula province, Samfya: single spore isolate from *Syzygium guineense*, 2008, D. Chungu (CMW29940/CBS124488)

*Chrysoporthe zambiensis*: Zambia, Luapula province, Kapweshi: single spore isolate from *Eucalyptus grandis*, 2008, D. Chungu (CMW29930/CBS124502)

*Chrysoporthe zambiensis* (CMW29930) and *Chrysoporthe syzygiicola* (CMW29940) isolates were obtained from the Culture Collection (CMW) of the Forestry and Agricultural Biotechnology Institute (FABI), Pretoria, South Africa. A phenol-choloroform protocol [25] was used to extract total genomic DNA (gDNA) from 14-day-old mycelium of these isolates grown in 2% (*w*/*v*) malt extract broth (Biolab, Merck, South Africa). High molecular weight gDNA was submitted for long read sequencing using a Pacific Biosciences Single-Molecule Real-Time (SMRT) protocol at Inqaba Biotechnical Industries (Pty) Ltd, Pretoria, South Africa. Furthermore, FastQC implemented in the Galaxy platform was used to evaluate the quality of the raw reads [26]. In addition, Canu was used to assemble the genomes [27], and QUAST [28] was used to determine general genomic statistics, such as N50, L50, and GC content, for the two isolates. BUSCO (benchmarking universal single-copy orthologs) [29] was used to evaluate the completeness of the draft genomes against the “sordariomycetes” database. Lastly, the AUGUSTUS de novo protein-coding gene prediction software [30,31] was used to annotate the draft genomes using gene models from *Neurospora crassa* as a reference set.

To confirm the taxonomic identities of the *C. zambiensis* (CMW29930) and *C. syzygiicola* (CMW29940) ex-type isolates, a phylogenomics approach was used. The draft genomes of the two isolates, along with previously sequenced genomes of other *Chrysoporthe* spp. were subjected to BUSCO analyses. A Python 3.8 command line script was used to parse the BUSCO output files and identify the genes that were complete and shared between all species in the analysis. The translated amino acid sequences were aligned using MUSCLE v. 5 [32,33], followed by automatic in silico trimming of each alignment using TrimAl v. 1.2 [34]. Amino acid alignments were concatenated to form a supermatrix, subjected to maximum-likelihood analysis using IQ-TREE v. 1.6.12 [35]. Confidence values in nodes were assessed using 1000 bootstrap replicates.

### 2.2. Structure of the Mating-Type Loci of C. zambiensis and C. syzygiicola

The draft genomes of *C. zambiensis* and *C. syzygiicola* were used to characterize their mating-type loci. The publicly available protein sequences for the *MAT1* gene models of *Cryphonectria parasitica*, namely *MAT1-1-1* (AAK83346.1), *MAT1-1-2* (AAK83345.1), *MAT1-1-3* (AAK83344.1), and *MAT1-2-1* (AAK83343.1) were retrieved from the National Center of Biotechnology Information (NCBI) GenBank database using their accession numbers. These sequences were used as query sequences against contigs of the draft genomes of *C. zambiensis* and *C. syzygiicola*. tBLASTn searches [36,37] were performed using CLC Main Workbench v.20.0 (CLC Bio, Aarhus, Denmark) to search for homologs of the *Cry. parasitica* mating-type genes in the *Chrysoporthe* genomes. In addition, tBLASTn searches were used to identify genes normally associated with the fungal *MAT1* locus [11], including the *APN2* (NCBI accession number: VM1G_08163), *COX6A* (NCBI accession number: VM1G_08162), and *APC5* genes [38]. Only sequences with at least 50% query coverage and contigs that produced matches with an E-value ≤0.01 were considered as possible homologs of the *MAT1* genes or genes associated with the flanking regions of the *MAT1* locus.

To annotate the contigs that putatively contain *MAT1* genes and its flanking genes, contigs were subjected to de novo gene prediction using the web-based AUGUSTUS gene prediction software [30,31], with the corresponding gene models from *Neurospora crassa* as references. BLASTp with default parameters were used to functionally characterize the predicted protein sequences of the putative *MAT1* genes, as well as genes associated with the *MAT1* locus, using the NCBI GenBank database. The conserved domains canonically associated with the mating-type genes were confirmed using the InterPro protein database [39,40] and Pfam [41].

To understand the structural differences and similarities of the *MAT1* loci of *Chrysoporthe*, the structures of the *MAT1* loci of previously published species [20] were reconstructed from available complete genome sequences. These locus structures were mapped onto the phylogenomic tree.

### 2.3. Phylogenetic Analysis of Mating-Type Genes

Maximum-likelihood gene trees of the mating-type genes were generated using IQ-TREE v. 1.6.12 [35], using the built-in selection of the best evolutionary model. These analyses included mating-type gene sequences of *C. puriensis* [16,42], *C. austroafricana* [43], *C. cubensis*, and *C. deuterocubensis* [44], with *Cry. parasitica* [22] as an outgroup taxon. MAFFT v. 7.1 [45] was used to perform multiple sequence alignments of mating-type genes and the combined dataset was visualized using the CLC Main Workbench v. 23, where poorly aligned regions were trimmed. The gene trees were compared to the species phylogeny generated in this study in order to detect incongruence.

To determine whether the mating-type genes from the genomes of *C. syzygiicola* and *C. zambiensis* were conserved, their inferred amino acid sequences were amended with homologous sequences from *C. puriensis*, *C. cubensis*, *C. deuterocubensis*, and *Cry. parasitica*. These amino acid sequences were aligned using MAFFT, and sequence similarities were inferred using the “create pairwise comparison” tool implemented in the CLC Genomics Workbench v. 23.

## 3. Results

### 3.1. Genome Sequencing and Taxonomic Confirmation

Both genomes sequenced during this study were submitted to NCBI under BioProject PRJNA971112. The *Chrysoporthe zambiensis* draft genome size was 48,317,394 bp (48.3 Mb), comprising 211 contigs. The L50 and N50 of the assembled genome were 19 and 691,378 bp, respectively, (Table 1). Moreover, the predicted number of gene models for *C. zambiensis* was 15,899, and BUSCO predicted 96.2% completeness for this genome (Table 2). For *C. syzygiicola*, the draft genome size was 42,500,337 bp (42.5 Mb), comprising 233 contigs. The L50 and N50 statistics of the *C. syzygiicola* draft genome were 21 and 617,420 bp, respectively, (Table 1). AUGUSTUS predicted 12,328 gene models, and BUSCO predicted a 95% genome completeness (Table 2).

When compared to other species of *Chrysoporthe*, namely *C. austroafricana* (44.67 Mb, 13,484 gene models), *C. cubensis* (42.62 Mb, 13,121 gene models), *C. deuterocubensis* (43.97 Mb, 13,772 gene models), and *C. puriensis* (44.66 Mb, 13,166 gene models), the genome size and predicted gene models of *C. zambiensis* were slightly larger, while the genome size and predicted gene models for *C. syzygiicola* were slightly smaller [42,43,44].

Phylogenetic analysis of *C. zambiensis* (CMW29930) and *C. syzygiicola* (CMW29940) using single-copy orthologs obtained from BUSCO analyses confirmed the identity of the isolates with 100% bootstrap values at all internal nodes (Figure 1).

### 3.2. Mating-Type Genes and Structure of the Mating-Type Loci of C. syzygiicola and C. zambiensis

The tBLASTn search against the whole-genome assembly of *C. syzygiicola* revealed the presence of *MAT1* genes and genes associated with the flanking regions of the *MAT1* locus on the same contig. The *MAT1-1-1*, *MAT1-1-2*, *MAT1-2-1*, DNA lyase (*APN2*), anaphase promoting complex (*APC5*), and cytochrome C oxidase subunit 6A (*COX6A*) genes were identified on contig ctg-000040F of the draft genome. AUGUSTUS predicted an additional five genes positioned between the *MAT1* genes, placing these within the mating-type locus of *C. syzygiicola* (Figure 2A). The genes located within the *MAT1* locus did not show any sequence similarities to any of the proteins present in the NCBI GenBank database, and no domains were detected from the protein databases, such as InterPro and Pfam.

tBLASTn searches against the draft genome assembly of *C. zambiensis* revealed the presence of the *MAT1-1-1*, *MAT1-2-1*, and *APN2* genes on a single contig (ctg000166F), while the *MAT1-1-2* gene was identified on a separate contig (ctg000072F). The genes associated with the flanking regions of the mating-type loci of fungal species [11,46] were not linked to the mating-type locus of *C. zambiensis*. In addition to the *MAT1* genes, seven genes with no known functions were present within the *MAT1* locus of *C. zambiensis*. None of these genes showed any amino acid sequence similarities with proteins from the GenBank database, and no domains were detected from the protein databases (Figure 2B).

In the mating-type locus of *C. syzygiicola*, the putative *MAT1-1-1* gene was 1098 bp long (CDS 1096 bp), coding 387 amino acids with no intron. In addition, this *MAT1-1-1* gene encoded a protein with the alpha-box domain (IPR006856) [10,11]. The putative *MAT1-1-2* gene was 1279 bp long (CDS 1119 bp) containing two introns (90 bp and 67 bp). The predicted *MAT1-1-2* gene encoded 373 amino acids, and no conserved motifs were detected against the Interpro and Pfam protein domain databases. The putative *MAT1-2-1* gene was 1008 bp long (CDS 879 bp), coding for 293 amino acids containing two introns (60 bp and 69 bp). The expected HMG box domain (IPR009071) that characterizes the *MAT1-2-1* gene was detected in both the Pfam and InterPro protein databases. The size of the mating-type locus was 25.25 kb. Moreover, based on the observed genetic composition of the mating-type locus of *C. syzygiicola*, this species has a homothallic mating system. Therefore, it is self-fertile and can complete sexual reproduction in the absence of a mating-type partner.

In the mating-type locus of *C. zambiensis*, the predicted *MAT1-1-1* gene was 1161 bp long (CDS 1161 bp), coding 387 amino acids that harbour the characterizing domain, the HMG box (IPR006856), with no introns detected in this gene. The putative *MAT1-1-2* gene was 1280 bp long, with a CDS of 1119 bp and two introns of 71 bp and 90 bp. The predicted *MAT1-1-2* gene encodes 373 amino acids, and no conserved motifs were detected against the Interpro and Pfam protein domain databases. The predicted *MAT1-2-1* gene was 1008 bp long with a CDS of 879 bp and two introns (60 bp and 69 bp). The *MAT1-2-1* gene encodes 293 amino acids and the HMG box domain (IPR009071), that characterizes this gene, was detected in both the Pfam and InterPro protein databases. The size of the *MAT1* locus of *C. zambiensis* was 28.97 kb. Based on the genetic content of the mating-type locus of this species, *C. zambiensis* is also homothallic.

The structures of the mating-type loci of *C. syzygiicola* and *C. zambiensis* were compared with the structures of the mating-type loci of other *Chrysoporthe* spp. using the species tree generated in this study (Figure 3). Based on this structural comparison, the genetic content of the *MAT1* loci of *C. syzygiicola* and *C. zambiensis* differed slightly from the genetic content of other *Chrysoporthe* spp. For example, the *MAT1* loci of homothallic *Chrysoporthe* spp. consist of genes that are associated with both the *MAT1-1* and *MAT1-2* idiomorphs, including *MAT1-1-1*, *MAT1-1-2*, *MAT1-1-3*, and *MAT1-2-1* genes. The *MAT1* loci of *C. zambiensis* and *C. syzygiicola* contain gene sequences for *MAT1-1-1*, *MAT1-1-2*, and *MAT1-2-1* that are homologous to *MAT1-1* and *MAT1-2* idiomorphs, but the *MAT1-1-3* gene is absent in the mating-type loci of both species. Additionally, genes associated with the flanking regions of the mating-type loci of Pezizomycotina, such as *COX6A* and *APC5*, were absent in the mating-type locus of *C. zambiensis*. The structures of mating-type loci of *Chrysoporthe* spp. are thus unique among the *MAT1* loci of filamentous ascomycetes.

### 3.3. Phylogenetic Analysis of the Mating-Type Genes

Sequence similarity comparisons (Table 3) of *C. syzygiicola* and *C. zambiensis* core *MAT1* genes against other species of *Chrysoporthe* and *Cry. parasitica* showed that the *MAT1-1-2* and *MAT1-2-1* genes tend to be more conserved among species than the *MAT1-1-1* gene. The core *MAT1* genes from *C. syzygiicola* also tended to be better conserved than those from *C. zambiensis*. Additionally, the average species-wise similarity values tended to decrease as relatedness decreased.

The maximum-likelihood gene trees generated for the *MAT1-1-1*, *MAT1-1-2* and *MAT1-2-1* core mating-type genes were incongruent with both each other and the species tree (Figure 4). The gene tree for *MAT1-1-1* was largely congruent with the species tree. However, in both the *MAT1-1-2* and *MAT1-2-1* gene trees, *C. zambiensis* and *C. syzygiicola* were more basal and grouped together with *C. austroafricana*. Species that were closely related to each other also displayed different mating systems. For example, while *C. zambiensis* and *C. syzygiicola* are homothallic, *C. austroafricana* is heterothallic. Therefore, the ancestral state for the mating system remains elusive.

## 4. Discussion

The genome sequences for *C. syzygiicola* and *C. zambiensis* allowed for the identification and characterization of the mating-type loci of these species, which in turn allowed inference regarding their mating systems. This study indicated that the *MAT1* loci of *C. zambiensis* and *C. syzygiicola* contain genes that are characteristic of homothallic mating systems, where *MAT1-1* and *MAT1-2* genes co-occur in the same genome [2,4]. The *MAT1* loci of both species harboured the *MAT1-1-1*, *MAT1-1-2*, and *MAT1-2-1* core mating-type genes. Apart from the mating-type genes, the *MAT1* loci of filamentous ascomycetes are usually associated with other non-mating-type genes, such as *APN2*, *COX6A*, and *APC5*, that are located in the flanking regions [11,47,48]. These genes were associated with the *MAT1* locus of *C. syzygiicola*, but *COX6A* and *APC5* were not associated with the *MAT1* locus of *C. zambiensis*.

Generally, the mating-type loci of homothallic Sordariomycetes harbour the *MAT1-1-1*, *MAT1-1-2*, *MAT1-1-3*, and *MAT1-2-1* genes [1,49,50]. However, the *MAT1-1-3* gene was absent in the *MAT1* loci of *C. syzygiicola* and *C. zambiensis*. Although the *MAT1-1-3* gene is frequently found in the *MAT1* loci of Diaporthales spp. and other fungi [1,20,22,51,52], but it can also be absent in some heterothallic [53,54] and homothallic species [55,56]. To date, the significance of the presence or absence of the *MAT1-1-3* gene in the *MAT1* loci of species of *Chrysoporthe* is unclear. However, in other fungi, specifically *Villosiclava virens*, the *MAT1-1-3* gene is essential for pathogenicity, sexual development, and asexual reproduction [57].

The predicted mating-type genes in the *MAT1* loci of *C. zambiensis* and *C. syzygiicola* have high sequence identity when compared with the *MAT1* genes of other *Chrysoporthe* species. The sizes of the *MAT1-1-1* genes of *Chrysoporthe* spp. from Zambia were similar, and the alpha-1 domain that characterizes this gene was also present. However, no intron was observed in the *MAT1-1-1* genes of *C. syzygiicola* and *C. zambiensis*. This trait seems to be unique among the Diaporthales, including *Chrysoporthe* spp. The sizes of the *MAT1-1-1* and *MAT1-1-2* genes of *C. syzygiicola* and *C. zambiensis* were slightly smaller in comparison to the same genes of *C. austroafricana*, *C. cubensis*, and *C. deuterocubensis* [20]. Furthermore, no conserved domain was observed in the *MAT1-1-2* genes of *C. syzygiicola* and *C. zambiensis*, a trait seemingly unique in these species. The gene and intron sizes of the *MAT1-2-1* genes of *C. syzygiicola* and *C. zambiensis* were conserved and similar to that of other *Chrysoporthe* spp. [20]. The presence of unknown genes within the mating-type loci of *C. zambiensis* and *C. syzygiicola* also a seemingly common feature that occurs in the *MAT1* loci of *Chrysoporthe* spp. [20].

The sizes of the mating-type loci in *Chrysoporthe* appear to be species-specific. For example, the size of the mating-type locus of *C. zambiensis* (29.0 kb) is larger than the *MAT1* loci of *C. syzygiicola* (20.9 kb) and *C. deuterocubensis* (18.2 kb). Similarly, the size of the *MAT1* locus of *C. cubensis* (45.0 kb) and the *MAT1-1* (133.8 kb) idiomorph of *C. austroafricana* were larger when compared to other *Chrysoporthe* spp. The mating-type loci in *Chrysoporthe* were consistently larger than the expected *MAT1* locus size of most filamentous ascomycetes studied here. The size variations of the *MAT1* loci of *Chrysoporthe* spp. might be attributable to the presence of varying numbers of genes of unknown function and the presence of transposable elements, observed in other *Chrysoporthe* spp. [20]. These transposable elements are associated with the expansion of the *MAT1* locus, introducing genetic variation, and suppressing recombination in this region if sexual reproduction is possible [46,58].

Compared to a typical sordariomycete *MAT1* locus, the structure of the *MAT1* locus of *Chrysoporthe* spp. is unique [1,2,11]. For example, a gene associated with the flanking regions of *MAT1* loci, such as *APN2* (AP endonuclease) [46,47,48,59], is present within the *MAT1* loci of all *Chrysoporthe* spp. studied thus far. Gene organization in the *MAT1* loci of *C. zambiensis* and *C. syzygiicola* is similar to what has been observed in other *Chrysoporthe* spp. [20]. However, the *COX6A* and *APC5* genes are located within the *MAT1* locus of *C. zambiensis*, instead of flanking it, when compared to the *MAT1* loci of other *Chrysoporthe* spp. Therefore, the structural configurations of the *MAT1* loci in the genus *Chrysoporthe* differ from each other and other ascomycetes.

In many filamentous ascomycetes, the structural configuration of the *MAT1* locus is *SLA2*—*MAT1*—*APN2*/*COX3A*/*APC5* [11] and the presence of these genes adjacent to the *MAT1* locus plays a crucial role in the identification and characterization of the *MAT1* locus. However, in some species of Diaporthales, the structural configuration of the *MAT1* locus is distinct from other filamentous ascomycetes. For example, in the *MAT1* locus of *Valsa mali* (Valsaceae, Diaporthales), *APN2* and *COX13* genes are located within the locus [38]. Additionally, the *APN2* gene is located within the *MAT1* loci of *Chrysoporthe* spp. The significance of the rearrangements of the *MAT1* loci is currently a matter of speculation. However, these rearrangements might induce beneficial genetic changes that can be selected for [58], thus potentially playing an important role in species evolution.

Incongruence between individual genes among each other and to a species tree has been observed before in Cryphonectriaceae [60], and in other fungi it has been linked to the speciation process [61]. Based on these observations, as well as tempting structural variation in the mitochondrial genomes of *Chrysoporthe* spp. [62], we can infer that divergence in this fungi group was likely driven by ancestral hybrid speciation, coincident with large-scale introgression. However, this conjecture remains to be tested and requires genome sequences from many more closely related species in the Cryphonectriaceae.

In this study, the homothallic mating systems of *C. syzygiicola* and *C. zambiensis* were confirmed. Based on the current analyses, there was no evidence of another mode of homothallism, such as mating-type switching, pseudohomothallism, or unidirectional mating [4], in *Chrysoporthe* spp. The characterization of the mating-type genes in these two species from Zambia indicate that they can reproduce sexually. However, the absence of perithecia in natural habitats could indicate that cryptic sex is taking place. In some fungal pathogens, the process of sexual reproduction and the presence of mating-type genes is associated with virulence [57,63]. Therefore, functional studies investigating the mating-type genes of *Chrysoporthe* spp. will be useful to understand the role these genes have in virulence, as well as the significance of the absent *MAT1-1-3* gene in the genomes of *C. zambiensis* and *C. syzygiicola*.

## Figures and Tables

**Figure 1 genes-14-01158-f001:**
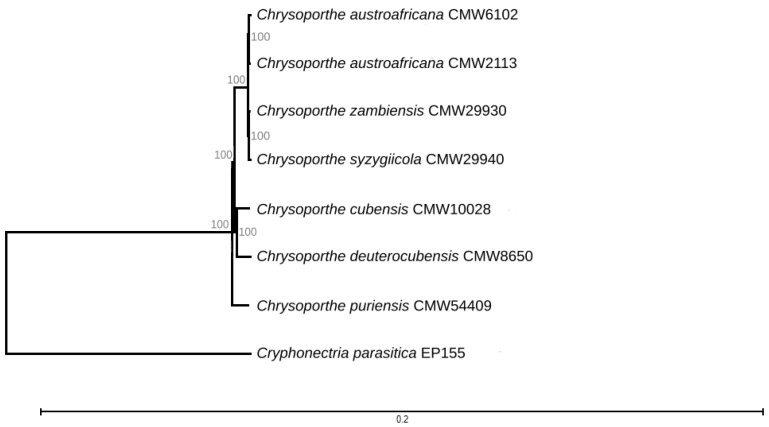
A maximum-likelihood tree generated from the combined BUSCO protein sequences of 3490 shared complete BUSCO genes from genomes of *Chrysoporthe* spp. Percentages at nodes denote bootstrap values (1000 replicates), while *Cryphonectria parasitica* EP155 was used as an outgroup taxon.

**Figure 2 genes-14-01158-f002:**
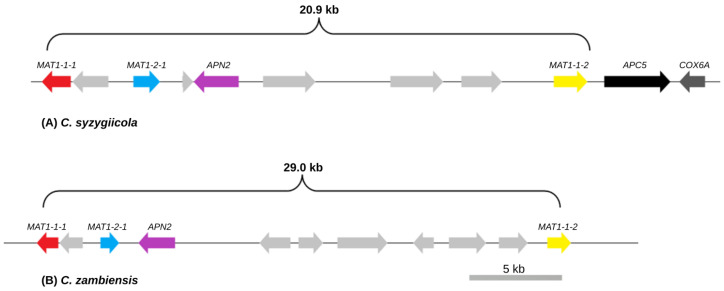
Scale diagram of the mating-type loci of (**A**) *C. syzygiicola* and (**B**) *C. zambiensis*. Genes coloured in light grey are those with unknown function or no known sequence similarities with genes from the NCBI database.

**Figure 3 genes-14-01158-f003:**
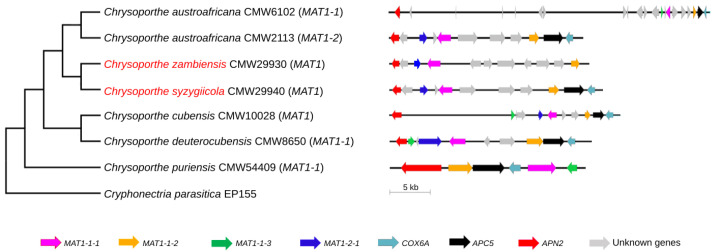
Structural comparison of the *MAT1* loci of *Chrysoporthe* spp., mapped onto the previously generated phylogenomic tree (Figure 1). Species highlighted in red are those sequenced and characterized in this study. *MAT1-1* and *MAT1-2* denote the idiomorphs of heterothallic species, while *MAT1* denotes a homothallic *MAT1* locus.

**Figure 4 genes-14-01158-f004:**
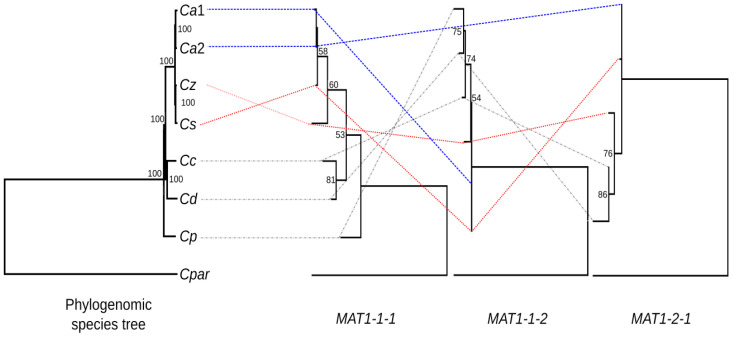
Phylogenetic incongruence between the species tree and maximum-likelihood gene trees for *MAT1-1-1*, *MAT1-1-2*, and *MAT1-2-1*. Red connectors denote the phylogenetic positions of the species from Zambia, while blue connectors denote *C. austroafricana* from South Africa. Bootstrap values above 50% (1000 replicates) are indicated at their respective nodes. Note that the *MAT1-1-1* sequence for *Ca*2 is a partial gene that is present in a *MAT1-2* idiomorph. *Cryphonectria parasitica* isolate EP155 (*MAT1-2*) was used as an outgroup for the phylogenomic tree and the *MAT1-2-1* gene tree, while isolate OB5-35 (*MAT1-1*) was used for the gene trees of *MAT1-1-1* and *MAT1-1-2*. *Ca*1: *Chrysoporthe austroafricana* CMW6102 (*MAT1-1*); *Ca*2: *Chrysoporthe austroafricana* CMW2113 (*MAT1-2*); *Cz*: *Chrysoporthe zambiensis* CMW29930 (homothallic); *Cs*: *Chrysoporthe syzygiicola* CMW29940 (homothallic); *Cc*: *Chrysoporthe cubensis* CMW10028 (homothallic); *Cd*: *Chrysoporthe deuterocubensis* CMW8650 (homothallic); *Cp*: *Chrysoporthe puriensis* CMW54409 (*MAT1-1*).

**Table 1 genes-14-01158-t001:** Genome assembly statistics for the draft genomes of *C. zambiensis* and *C. syzygiicola*.

Assembly Metric	*C. zambiensis*	*C. syzygiicola*
Genome size (bp)	48,317,394	42,500,337
Number of contigs	211	233
GC content (%)	56.57	55.43
N50 (bp)	691,378	617,420
L50	19	21

**Table 2 genes-14-01158-t002:** Completeness statistics for the draft genomes of *C. zambiensis* and *C. syzygiicola*.

BUSCO Statistic	*C. zambiensis*	*C. syzygiicola*
Overall completeness (%)	96.2	95
Complete BUSCO genes (C)	3672	3631
Single copy orthologs (S)	3664	3625
Duplicated orthologs (D)	8	6
Fragmented orthologs (F)	34	39
Missing orthologs (M)	111	147

**Table 3 genes-14-01158-t003:** Similarity comparison of the coding sequences of the mating-type genes of *C. syzygiicola* and *C. zambiensis* in relation to other species of *Chrysoporthe* and *Cry. parasitica*. The *MAT1-1-1* gene in the *MAT1-2* idiomorph of *C. austroafricana* is truncated. All numerical values are percentages. Entries indicated with “—” denote self-comparisons, or where a gene from a homothallic species did not exist in an idiomorph of a heterothallic species.

		NCBI	*C. syzygiicola*				*C. zambiensis*			
Species	Isolate	Assembly	*MAT1-1-1*	*MAT1-1-2*	*MAT1-2-1*	AVG	*MAT1-1-1*	*MAT1-1-2*	*MAT1-2-1*	AVG
*C. syzygiicola*	CMW29940	PRJNA971112	—	—	—	—	91.29	98.11	97.12	95.50
*C. zambiensis*	CMW29930	PRJNA971112	91.29	98.11	97.12	95.50	—	—	—	—
*C. austroafricana*	CMW6102 (*MAT1-1*)	ASM105115	97.51	100	—	98.76	92.57	98.11	—	95.34
*C. austroafricana*	CMW2113 (*MAT1-2*)	ASM1607180	98.76	100	99.64	99.47	84.57	98.11	97.48	93.39
*C. cubensis*	CMW10028	ASM128231	84.65	97.57	97.12	93.11	87.55	96.22	97.48	93.75
*C. deuterocubensis*	CMW8650	ASM151382	87.55	96.76	94.24	92.85	83.55	95.41	94.60	91.19
*C. puriensis*	CMW54409 (*MAT1-1*)	ASM1567895	84.23	95.41	—	89.82	83.82	94.05	—	88.94
*Cry. parasitica*	EP155	Crypa2	46.47	52.15	62.23	53.62	45.23	51.61	61.87	52.90
Per gene average similarity			84.35	91.43	90.07		81.23	90.23	89.71	

## Data Availability

Draft genome sequences generated during this study were deposited at the NCBI under BioProject PRJNA971112 https://www.ncbi.nlm.nih.gov/bioproject/?term=PRJNA971112 (accessed on 19 May 2023).

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
