# Peer review of "The Unique Homothallic Mating-Type Loci of the Fungal Tree Pathogens Chrysoporthe syzygiicola and Chrysoporthe zambiensis from Africa"

_genes, 2023, doi:10.3390/genes14061158_

Round 1

Reviewer 1 Report

In the manuscript ‘The homothallic mating-type loci of the fungal tree pathogens Chrysoporthe syzygiicola and Chrysoporthe zambiensis from Africa’, the authors sequenced two African Chrysoporthe sp, and performed a set of analysis of MAT loci, concluding that these both species have homotallic trait of sexual reproduction.

In my opinion, presented work is interesting and it is important to the other fungal researchers working in the fungal sexual reproduction, especially regarding plant pathogenic fungi. The carefully thought-out experiments gave the interesting results. I see few points concerning this work, which may improve the manuscript after small revision:

The particular mechanism and the genes/proteins involved in mating (MAT, APC5, COX6A, APN2) should be breathily described in the introduction to clarify this mechanism. Also, the differences between the MAT loci.

Row 97: this section 2.2. may be moved and sticked to section 2.1.

Row 155: When compared to other species... which ones? Next...were slightly smaller-than what? Any numbers?

The same, table 2: BUSCO statistics are related to what?

Row 279: ..of Chrysoporthe is unique – and this is the main scope! Should be emphasized in the introduction and the title

Author Response

Reviewer point 1: The particular mechanism and the genes/proteins involved in mating (MAT, APC5, COX6A, APN2) should be breathily described in the introduction to clarify this mechanism. Also, the differences between the MAT loci.

Response 1: We believe that we have provided adequate detail for this study in the second paragraph of the Introduction. Additionally, given the extreme variability and complexity of the system, which has been adequately reviewed before (and cited in our manuscript), we are of the opinion that further detail included in the text might detract from the main point of the research reported here.

Reviewer point 2: section 2.2. may be moved and sticked to section 2.1.

Response 2: We agree and have prepended this section to the text of Section 2.1

Reviewer point 3: The Reviewer asked for more detailed information for the comparison of genome sizes and numbers of gene models between the isolates sequenced in this study and those published in the literature (original submission line 155).

Response 3: We explicitly provided this information in the text.

Reviewer point 4: Table 2 reported on the BUSCO statistics of the two genomes sequenced in this study. Reviewer 1 wanted to know what these statistics are related to.

Response 4: BUSCO statistics are not reported in relation to other genomes, since these statistics are free-standing scalars (i.e., magnitude only) and not vectors (magnitude and direction). The BUSCO software uses precomputed HMM models to identify a set of genes in a genome. In this case, the HMMs were derived from a class of ascomycetes known as sordariomycetes, which is commonly used for the Cryphonectriaceae and allows comparisons between studies. This information was indicated in the Materials and Methods of the original manuscript (Line 84).

Reviewer point 5: The Reviewer suggested that the uniqueness of the mating-type loci of C. syzygiicola and C. zambiensis should be emphasized.

Response 5: We edited the title of the manuscript to include the word "unique". We also added a sentence in the Introduction (Lines 48-49 of the corrected manuscript) to emphasize this point.

Reviewer 2 Report

Dear Authors, 

my comments are included in the main text. However, the crucial questions/comments are:

1. Are you sure that your strains were single-spore ones?

2. In Fig. 2, in the illustration of genes of C. zambiensis I can not see the symbol (dark blue) of the MAT1-2-1 indicating the homothallic state. Is it the oversight or a mistake, or true, that changes the result?

Author Response

Reviewer comment 1: Are you sure that your strains were single-spore ones?

Reply 1: the isolates sequenced in this study are single-conidium isolates. These isolates are also deposited at the Westerdijk Institute, The Netherlands (i.e., CBS numbers) as single spore ex-types. This is indicated under section 2.1

Reviewer comment 2: In Fig. 2, in the illustration of genes of C. zambiensis does not have the symbol (dark blue) for the MAT1-2-1 gene that will indicate the homothallic state. Is this an oversight, a mistake, or a true difference which will changes the results?

Reply 2: The problem with gene coloring in Figure 2 was actually in Figure 3. We have corrected the coloring oversight in Figure 3, while Figure 2 remains unchanged.

Review 3: The reviewer supplied an annotated PDF file of the manuscript with suggested edits and corrections.

Reply 3: All of these edits and corrections have been incorporated, including the missing year-dates in references 1-7. The only change not made relates to the captions of figures, specifically Figure 2, that should not end with a full stop. However, we disagree since the caption of Figure 2 ends in a full sentence.